# Physiological Responses of Basil (*Ocimum Basilicum* L.) Cultivars to *Rhizophagus Irregularis* Inoculation under Low Phosphorus Availability

**DOI:** 10.3390/plants9010014

**Published:** 2019-12-20

**Authors:** Boris Lazarević, Klaudija Carović-Stanko, Zlatko Šatović

**Affiliations:** 1Department of Plant Nutrition, Faculty of Agriculture, University of Zagreb, Svetošimunska cesta 25, 10000 Zagreb, Croatia; 2Centre of Excellence for Biodiversity and Molecular Plant Breeding, Svetošimunska cesta 25, 10000 Zagreb, Croatia; kcarovic@agr.hr (K.C.-S.); zsatovic@agr.hr (Z.Š.); 3Department of Seed Science and Technology, Faculty of Agriculture, University of Zagreb, Svetošimunska cesta 25, 10000 Zagreb, Croatia

**Keywords:** basil, chlorophyll fluorescence, gas exchange, mycorrhizal responsiveness, phosphorus nutrition, root morphology

## Abstract

Arbuscular mycorrhizas (AM) can improve phosphorus (P) nutrition and could serve as an environmentally friendly approach for sustainable crop production under P-limiting conditions. The objectives of this study were to assess the effect of AM on different physiological traits and to quantify the responsiveness of different basil (*Ocimum basilicum* L.) cultivars to AM under low P availability. The basil cultivars ‘Genovese’, ‘Sweet Basil’, ‘Dark Opal’, and ‘Erevanskii’ were inoculated (AMI) using *Rhizophagus irregularis*. Photochemical efficiency and gas exchange were measured on AMI and non-inoculated (AMC) plants and, at harvest, the shoot biomass, shoot P concentration, root morphological traits, frequency of mycorrhizas in the roots (F%), and extent of root colonization (M%) were determined. Significant differences in F% and M% were found among the examined cultivars, with the highest found in ‘Dark Opal‘ and the lowest in ‘Erevanskii‘. AMI reduced the shoot biomass and increased the shoot P concentration as well as other examined root traits in ‘Genovese’ and ‘Erevanskii’, whereas it did not affect those traits in ‘Dark Opal’ and ‘Sweet Basil’, indicating differences in responsiveness to AM. AMI positively affected the gas-exchange parameters in all examined cultivars, probably due to the increased sink capacity of a bigger root system and/or AM structures within the roots.

## 1. Introduction

Soil resource acquisition is a primary limitation on crop production [1], and phosphorus (P) is known to be the least available plant nutrient under most soil conditions [2]. Limited inorganic P supply affects numerous plant growth and physiological processes. Several authors have pointed out that phosphorus deficiency leads to a reduced rate of photosynthesis, by reducing the carboxylation efficiency and apparent quantum yield [3,4], sucrose synthesis and carbon partitioning [5], and shoot growth [6,7], and affecting root growth and architecture [8].

The traditional approach to overcome P limitation is soil fertilization with large amounts of P fertilizers. However, due to the high phosphate-binding capacity of soils, plants use only 10–25% of applied P from fertilizers [9]. In addition, the global decline in phosphate rock supplies as a non-renewable source represents a challenge for traditional agriculture, which requires alternative approaches for P fertilization strategies [10]. One of the widely spread adaptations to low P availability in terrestrial ecosystems, which increases P uptake by plants, is the development of mycorrhizal associations [11]. Thus, the use of arbuscular mycorrhizas (AM) can represent an environmentally friendly approach for sustainable crop production under P-limiting conditions.

Basil (*Ocimum basilicum* L.) is one of the most cultivated and used medicinal and aromatic plants and it has long been acclaimed for its diversity as a source of essential oils, its flavor and delicacy as spice, and its beauty and fragrance as an ornamental plant [12]. Within the species, there are five main botanical varieties: var. *basilicum* L., var. *difforme* Benth., var. minimum L., var. *purpurascens* Benth. and var. *thyrsiflorum* /L./Benth. [13]. The species also exhibits a variety of cultivars grown for various purposes. Carović-Stanko et al. [13] offered the classification of basil cultivars into six distinct morphotypes: Lettuce-leaf, Small-leaf, True basil, Purple basil (A), Purple basil (B) and Purple basil (C). Furthermore, Varga et al. [14] propose the intraspecific characterization of *O. basilicum* into five chemotypes: (A) High-linalool, (B) Linalool/trans-α-bergamotene, (C) Linalool/methyl chavicol, (D) Linalool/trans-methyl cinnamate and (E) High-methyl chavicol chemotype, based on the essential oil composition of 85 accessions.

Basil is considered a responsive species to arbuscular mycorrhiza (AM) inoculation [15]. Several reports have pointed out the positive effects of AM inoculation on basil nutrient uptake [16], essential oil synthesis [17], phenolic compound synthesis [15,16,18] and enhanced tolerance to increased salinity stress [19] and water-deficit stress [18]. However, the responsiveness of different plant species to AM colonization in terms of improved growth and mineral nutrition varies enormously among genotypes of each species [11] and there is scarce information about the influence of AM on the physiological performance of different basil cultivars under P-limited conditions.

In this research, some of the most widely grown cultivars [*O. basilicum* var. *basilicum* ‘Genovese’ (True basil morphotype and Linalool chemotype), *O. basilicum* var. *basilicum* ‘Sweet Basil’ (True basil morphotype and Linalool chemotype), *O. basilicum* var. *purpurascens* ‘Dark Opal’ (Purple basil (B) morphotype and Linalool chemotype) and *O. basilicum* var. *purpurascens* ‘Erevanskii’ (Purple basil (A) morphotype and Methyl chavicol chemotype)] were included.

The objectives of this study were: (i) to quantify the potential for colonization and responsiveness of different basil cultivars to AM inoculation, and (ii) to assess the effect of AM inoculation on plant biomass, phosphorus nutrition, root morphology, photochemical efficiency, and the gas-exchange parameters of different basil cultivars under low P availability.

## 2. Results

### 2.1. Chlorophyll Fluorescence

The differences among basil cultivars regarding the maximal quantum yield of PS 2 (Fv/Fm), the actual quantum yield of PS 2 (Y(II)), the apparent electron transport rate (ETR), and non-photochemical quenching (NPQ) were determined (Figure 1). In addition, there was significant cultivar × AM inoculation interaction for Fv/Fm, Y(II), ETR, and NPQ. Namely, compared to the control, AMI increased Fv/Fm, Y(II), and ETR only in ‘Genovese’ and ‘Dark Opal’, and decreased NPQ only in ‘Dark Opal’ (Figure 1).

### 2.2. Gas Exchange

The differences among basil cultivars regarding the net photosynthetic rate (A), stomatal conductance (g_s_), transpiration rate (E), and water-use efficiency (WUE) were determined (Figure 2). AMI increased the average A, g_s_, E, and WUE across all basil cultivars (Figure 2). However, E and WUE were affected by significant cultivar × AM inoculation treatment interaction since AMI increased E onlyin ‘Genovese’ and ‘Dark Opal’ (Figure 2d) and WUE only in ‘Sweet Basil’ and ‘Erevanskii’ (Figure 2e).

### 2.3. Shoot Biomass and Shoot Mineral Concentration

AMI decreased the average shoot dry weight (DW), on average by 14.3%, for all cultivars. In addition, there were differences in average shoot dry weight among basil cultivars, with highest found in ‘Dark Opal’ and lowest in ‘Genovese’ (Figure 3a). However, shoot dry weight was significantly affected by Cultivar × AM treatment interaction since AMI decreased shoot dry weight of ‘Genovese’ and ‘Erevanskii’ and did not significantly affect shoot dry weight of ‘Dark Opal’ and ‘Sweet Basil’ (Figure 3a).

Differences among basil cultivars were found for the average shoot P concentrations. The highest average P concentration (for both AMI and AMC plants) was found in ‘Erevanskii’ (1.45 mg g^−1^ DW) and the lowest in ‘Genovese’ (0.89 mg g^−1^ DW). In addition, the average shoot P concentrations were 1.3 mg g^−1^ DW and 1.03 mg g^−1^ DW for ‘Sweet Basil’ and ‘Dark Opal’, respectively (Figure 3b). AMI increased the average shoot P concentration in all basil cultivars (by 9.9%). However, there was significant cultivar × AM treatment interaction, since AMI increased the shoot P concentration only in ‘Genovese’ (19.8%) and ‘Erevanskii’ (13.7%) and did not affect the shoot P concentration in ‘Dark Opal’ and ‘Sweet Basil’ (Figure 3b).

A significant effect on the basil cultivars was found for calcium (Ca), magnesium (Mg), iron (Fe), and zinc (Zn) content in the shoot dry weight (DW) (Table 1). The highest average Ca content (1.53 mg g^−1^ DW) and lowest iron content (76.6 mg g^−1^ DW) was found in ‘Erevanskii’. The highest average Zn content was found in ‘Dark Opal’ (32.0 mg g^−1^ DW), whereas a higher Mg content was found in ‘Genovese’ (0.44 mg g^−1^ DW) and ‘Sweet Basil’ (0.45 mg g^−1^ DW) compared to ‘Dark Opal’ (0.31 mg g^−1^ DW) and ‘Erevanskii’ (0.33 mg g^−1^ DW). There was no significant effect of AMI treatment on the content of Ca, Mg, Fe, Zn and manganese (Mn) in the examined basil cultivars (Table 1).

### 2.4. Root Morphology

Differences among basil cultivars were found for all of the examined root traits. The largest root systems in respect to all measured traits apart from the average root diameter, across both AMI and AMC treatments, were found in ‘Genovese’ and ‘Erevanskii’ (Table 2). AMI increased the average root length by 60.7%, the surface area by 65.2%, the volume by 58.32%, the depth by 39.8%, the width by 22.7% and the number of tips by 88.5%, and it decreased the average root diameter by 7.9% (Table 2). However, significant cultivar × AM treatment interactions were identified due to a significant increase in root depth, length, surface area, volume and number of root tips in the AMI compared to AMC plants of ‘Genovese’ and ‘Erevanskii’, while these root traits did not differ between the AMI and AMC plants in ‘Sweet Basil’ and ‘Dark Opal’ (Table 2).

### 2.5. Arbuscular Mycorrhizas (AM) Root Colonization

The results of the root colonization analyses show that basil cultivars possess considerable affinity for the development of AM symbiosis. However, significant differences in the F% and M% were found among the studied cultivars. The highest F% (70.5%) and M% (54.0%) were found in ‘Dark Opal’ and the lowest in ‘Erevanskii’, 36.3% and 17.0%, respectively (Figure 4). There was no significant difference in M% between ‘Genovese’ and ‘Erevanskii’, and in F% between these two cultivars and ‘Sweet Basil’ (Figure 4a,b).

### 2.6. Mycorrhizal Responsiveness

The responsiveness of the examined basil cultivars to AM colonization was estimated for each cultivar by calculating the percent difference in the (i) shoot biomass (shoot dry weight), (ii) shoot P concentrations, and (iii) root length, between the AMI and AMC plants. AMI caused a decrease in the average shoot biomass and an increase in the average shoot P concentration and the average root length. However, the basil cultivars differed remarkably in the responsiveness to AM inoculation. Namely, the biggest biomass reduction caused by AMI was found in ‘Genovese’ (−32.8%) and ‘Erevanskii’ (−17.9%), whereas less biomass reduction was found in ‘Sweet Basil’ (−5.25%) and ‘Dark Opal’ (−1.35%) (Figure 5). The opposite results were found in the percent differences in root lengths, where the highest root length increase was found in ‘Erevanskii’ (68.3%) and ‘Genovese’ (60.2%) and the lowest in ‘Dark Opal’ (14.9%) and ‘Sweet Basil’ (8.3%) (Figure 5). Similarly, the P concentrations in the AMI plants were 19.8% and 13.7% higher compared to the AMC plants of ‘Genovese’ and ‘Erevanskii’, respectively. Meanwhile, the P concentrations in the AMI plants of ‘Dark Opal’ and ‘Sweet Basil’ were 3.4% and 6.7% higher compared to the AMC plants, respectively.

## 3. Discussion

Low phosphorus availability represents a major constraint to plant growth [1], whereas symbiosis with AM fungi represents the most widespread plant adaptation to low P availability [11] in terrestrial ecosystems. However, there is considerable variation in affinity and responsiveness to AM inoculation among different plant species, as well as among different genotypes of the same species [11]. The results of this study show that basil cultivars possess high potential for AM colonization under limited P availability, which is in line with the results obtained by [15]. In addition, significant differences existed in the frequency of mycorrhizas in the root system (F%) and in the extent of root cortex colonization (M%) among the examined basil cultivars, with the highest F% and M% being found in ‘Dark Opal’ and the lowest in ‘Erevanskii’ (Figure 4a,b). Besides genotypic differences, specific plant responsiveness to AM can differ due to inoculation with different AM species [20] and due to different soil P availability levels [21]. Although, in this experiment, single AM species and single P levels were used, differences in the development of AM structures within root systems and the specific responsiveness of each cultivar to *R. irregularis* inoculation under limited P nutrition had substantial effect on the examined morphological and physiological traits.

Namely, through the analysis of the improvement in shoot biomass by comparing the percent differences in shoot dry weight between AMI and AMC plants, it can be stated that the responsiveness of basil cultivars to *R. irregularis* varies from the negatively responsive cultivars, ‘Genovese’ and ‘Erevanskii’, to the neutral responsive cultivars, ‘Sweet Basil’ and ‘Dark Opal’ (Figure 3a; Figure 5). The opposite results were found in the responsiveness measured by the percent differences in the shoot P concentration, where ‘Erevanskii’ and ‘Genovese’ responded positively, whereas ‘Dark Opal’ and ‘Sweet Basil’ had a neutral response (Figure 5). These results indicate that the responsiveness of different genotypes to AM inoculation should be estimated by several indicators at the whole-plant level.

Although AM are known for their positive effect on the P uptake, the obtained increase in shoot P concentration in ‘Genovese’ and ‘Erevanskii’ could be more related to the significant decrease in shoot dry weight of AMI plants (Figure 3a) than to the increased P uptake efficiency caused by AM inoculation. In addition, a slight but non-significant increase in the shoot concentration of other mineral nutrients was found in the AMC plants compared to the AMI plants of those two cultivars (Table 1). In addition, the shoot biomass reduction in the AMI plants of ‘Genovese’ and ‘Erevanskii’ is probably related to the significant increase in the root system in AMI plants of these two cultivars (Table 2; Figure 5). A growing root system represents a significant sink for assimilated C and thus may cause a reduction in shoot biomass. These results are in line with the negative responsiveness of the ‘Genovese’ and ‘Erevanskii’ cultivars, because most research has shown that root growth decreases with AM inoculation, due to a change in the P uptake pathway (which can be up to 100% acquired by AM), thus the AM responsive plants reduce the C allocation to the roots, and decrease root growth [22,23]. In addition, responsiveness to AM may be associated with root traits, whereby plants with short root length, less branched roots, and larger diameter roots could be more responsive to AM [11]. Such root phenotypes are often described as less efficient in P acquisition from soil [1], and thus probably rely more on the uptake of P by the AM pathway. The fact that AM colonization caused increased root growth for ‘Genovese’ and ‘Erevanskii’ and did not affect those in ‘Sweet Basil’ and ‘Dark Opal’ could mean that AM colonization did not entirely modify the P uptake mechanism in the host plants of ‘Genovese’ and ‘Erevanskii’, and thus the plants of these cultivars responded to the P deficit by favouring root growth. Similar results were obtained by Liu et al. [24], who found that AM colonization decreased C and N partitioning in soybean stems and leaves, and increased C and N partitioning in the root system under water deficit stress. In addition, Feldmann et al. [25] stated that plant benefits from AM were not evident under the threshold of about 30% root colonization, which could explain the lack of a positive effect of AM inoculation on the ‘Genovese’ and ‘Erevanskii’ cultivars (Figure 4a,b).

The determined average shoot P concentrations (under 1.5 mg g^−1^ DW) were below those needed for optimal plant growth (3–5 mg g^−1^ DW) [26], indicating a P deficit in all examined cultivars in both AMC and AMI treatment. In addition, other studies have shown shoot P concentrations of basil plants grown under non-limiting P conditions substantially higher (6–11 mg g^−1^ DW) [27] compared to results of this study. Although significant differences were found in the Ca, Mg, Fe and Zn content among the examined basil cultivars, the results show that all plants were sufficiently supplied with these nutrients [26]. Such results are probably caused by a surplus supply of nutrients provided by frequent irrigation with a nutrient solution, and thus masking the possible positive effect of AM on nutrient uptake. Determined low shoot P concentrations can disturb photosynthesis, both the light reactions and carbon reactions [3,4,28]. However, carbon assimilation is important for sustaining AM symbiosis and fungal growth. In addition, there is evidence that AM inoculation can increase plant photochemical efficiency [19] and C assimilation rate [29,30,31] under different stressful factors. The results of this study show that AMI can increase photochemical efficiency in basil under P deficit stress. Namely, AMI increased the F_v_/F_m_, Y(II), and ETR in ‘Genovese’ and ‘Dark Opal’ (Figure 1). For the AMI plants of ‘Genovese’, the increased photochemical efficiency could be attributed to higher shoot P concentrations, whereas for ‘Dark Opal’ this is probably related to a decreased energy dissipation, measured by NPQ (Figure 1d). An even more obvious positive effect of AMI was found in the gas-exchange parameters, especially regarding stomatal conductance and net photosynthetic rate, which were higher for the AMI plants compared to the AMC plants of all cultivars (Figure 2a,c). As was shown by Ravnskov et al. [30] the increase in photosynthetic rate could be explained by the increased sink capacity of AMI plants due to increased needs for sustaining AM growth, and due to increased root growth in ‘Genovese’ and ‘Erevanskii’. However, as results of decreased shoot dry weight show, increased photosynthetic rate of AM inoculated Genovese’ and ‘Erevanskii’ plants was not enough to sustain both shoot and root growth. Moreover, Paul and Foyer [32] stated that photosynthetic control is responsive to the whole plant’s needs, thus changes in sink metabolism provide feedback regulation of the source activity, whereas Rychter and Rao [28] stated that sink strength imposes the most important regulatory role on photosynthesis during phosphate deficiency. In addition, the observed increase in g_s_ facilitates CO_2_ diffusion into the leaf mesophyll, which increases the carboxylation rate of the RuBisCO enzyme, and thus increases the rate of photosynthesis. Concomitantly with the increase in g_s_, an increase in E was obtained; however, as could be seen from the increase in WUE, AMI caused a higher increase in A compared to E (Figure 2e). AM’s effect on stomatal conductance is well documented and it is attributed to different factors, such as increased P nutrition [33], plant hormonal changes [34], bigger root surface area [35] and improved soil conductivity [36]. Although this experiment was conducted on a limited number of plants and can be considered as preliminary work and, furthermore, these findings should be confirmed on a bigger sample, the results of this experiment show P deficiency levels in all the examined basil plants, and it seems that the increased photochemical efficiency, stomatal conductivity, net photosynthetic rate, transpiration rate and water-use efficiency could be explained only as a whole-plant response to AM inoculation, which includes increased sink capacity of the roots and/or AM structures, as well as hormonal changes and the soil/substrate changes in the rhizosphere.

## 4. Conclusions

AM symbiosis represents the most widespread plant adaptation to low P availability in terrestrial ecosystems. The examined basil cultivars differed in responsiveness to AM, which was either negative, where AM caused a shoot biomass reduction (‘Genovese’ and ‘Erevanskii’), or neutral (‘Dark Opal’ and ‘Sweet Basil’). Under P-limiting conditions, the neutral responsive cultivars supported the development of AM structures within the root (showing higher M% and F%), whereas the negatively responsive cultivars invested more in root growth. An increased sink capacity caused positive feedback regulation, increasing photochemical efficiency and gas-exchange parameters in AM-inoculated basil plants.

## 5. Materials and Methods

### 5.1. Plant Material and Experimental Conditions

Four basil accessions from the Collection of Medicinal and Aromatic Plants of the Department of Seed Science and Technology, University of Zagreb Faculty of Agriculture, Croatia (http://cpgrd.hcphs.hr); (MAP00145 ‘Genovese’, MAP00232 ‘Sweet Basil’, MAP00333 ‘Dark Opal’, MAP00335 ‘Erevanskii’) were analysed.

First, the seeds were surface sterilised using 1% sodium hypochlorite for 10 min and rinsed three times under distilled water. The plants were grown in a growth chamber for 60 days in 2L plastic pots filled with vermiculite (Gramoflor Vermiculite, Gramoflor GmbH & Co, Vechta, Germany) at 300 µmol m^−2^ s^−1^ PAR, 25/20 °C, 16/8 h day/night period, and 75% relative air humidity. Prior to planting, the substrate was irrigated with distilled water to 60% of the field capacity. Chemical analysis of the used vermiculite was performed in an analytical laboratory in the Department of Plant Nutrition, Faculty of Agriculture, University of Zagreb. Analysis of electroconductivity (EC) (Mettler Toledo Seven Compact, pH and conductivity meter) showed that it contained small amounts of water-soluble salts (0.038 mS cm^−1^), thus the total nutrient content was analysed after the combustion of the material in an Ethos UP microwave digestion system (Milestone Srl, Italy) in a mixture of hydrochloric and nitric acid (3:1). The total phosphorus content was determined spectrophotometrically (Thermo Scientific™ Evolution™ 60S UV-Visible Spectrophotometer), the potassium content was determined by flame photometer (Jenway PFP7, Flame Photometer, Staffordshire UK), while the calcium and magnesium content were determined by atomic absorption spectroscopy (Thermo Scientific - SOLAAR M Series AA Spectrometer) [37]. The basic properties of the used vermiculite were as follows: 6.96 pH, 0.8% P_2_O_5_, 3.1% K_2_O, 6.2% CaO, 22.0% MgO, and 115 g L^−1^ bulk density.

The experiment was set as a randomized complete block design and each cultivar was represented with 4 plants per treatment (one plant per pot). Arbuscular mycorrhiza (AM) inoculation treatments were represented as AM-inoculated (AMI) and the control, i.e., the non-inoculated (AMC) plants. AM inoculation was performed by the application of 0.5 g (equivalent of 2000 spores) of Mycodrip (*Rhizophagus irregularis*, Symbiom LTD) per pot during sowing. The AMC plants received 0.5 g of spore-free diatomaceous earth, which is used as a carrier substrate for spores in Mycodrip. Once a week, the plants were irrigated with 200 mL of a nutrient solution (pH 6.0) containing: KCl 1 mM; NH_4_NO_3_ 1.5 mM; CaCl 1 mM; KH_2_PO_4_ 3 µM; MgSO_4_ 200 µM; Mg(NO_3_)_2_ 500 µM; MgCl_2_ 155 µM; MnCl_2_ x 4H_2_O 11.8 µM; H_3_BO_3_ 33 µM; ZnSO x 7H_2_O 3.06 µM; CuSO_4_ x 5H_2_O 0.8 µM; Na_2_MoO4H_2_O 1.07 µM; Fe-HEDTA 77 µM. Thus, by eight irrigations, a total amount of 0.653 mg of KH_2_PO_4_ was applied to each plant. The supplemental irrigations were performed by distilled water to keep the substrate at 50−60% of field capacity, monitored by W.E.T. Sensor Kit (Delta-T Devices LTD., Cambridge, UK).

### 5.2. Chlorophyll Fluorescence and Gas-Exchange Measurements

The chlorophyll fluorescence and gas-exchange measurements were performed 50 days after planting (sowing), using the Plant Stress Kit (Opti-Sciences, Inc. Hudson USA) and the LCpro portable photosynthesis system (ADC, Bio Scientific Ltd., UK), respectively. The plants were irrigated the day before measurements. Three measurements per leaf blade of the youngest fully developed leaf were recorded and the average value per plant was calculated to compensate for the leaf heterogeneity.

The minimal fluorescence (F_0_) was measured using low light flesh (0.15 μmol m^−2^ s^−1^ PAR) and the maximal fluorescence (F_m_) was measured using saturation pulse (5000 μmol m^−2^ s^−1^ PAR) during 1 s in dark-adapted leaves (overnight dark adaptation e.g., 8:00 h of dark period). In light-adapted leaves (after 3 h of actinic light, 300 μmol m^−2^ s^−1^ PAR) under steady-state photosynthesis, the steady-state fluorescence (F_s_) and the maximal fluorescence (F_m’_) using saturation pulse (5000 μmol m^−2^ s^−1^ PAR) during 1 s were measured. Based on the chlorophyll fluorescence measurements, the maximal quantum yield of PS 2 (F_v_/F_m_), actual quantum yield of PS 2 (Y(II)), apparent electron transport rate (ETR), and non-photochemical quenching (NPQ) were calculated using the following formulas [38,39]:F_v_/F_m_ = (F_m_ − F_0_) / F_m_,(1)
Y(II) = (F_m_’ − F_s_)/F_m_’,(2)
ETR = Y(II) × PAR × 0.84 × 0.5,(3)
NPQ = (F_m_ − F_m_’)/F_m_’,(4)

For the ETR estimation, the average leaf light absorbance of 84%, and the portion of light provided to PS 2 of 50% was assumed.

The leaf gas-exchange parameters [net photosynthetic rate (A), transpiration rate (E), stomatal conductance (g_s_), and intercellular CO_2_ concentration (C_i_)] were measured at 1200 μmol m^−2^ s^−1^ photosynthetically active radiation (PAR) and at 400 ± 5 μmol mol^−1^ CO_2_ concentration.

The instantaneous water-use efficiency (WUE) and the intrinsic water use efficiency (WUE_i_) were calculated as A/E and A/g_s_, respectively.

### 5.3. Shoot Biomass and Mineral Concentration Analysis

At harvest, 60 days after planting, the shoots and roots were separated. Before DW determination, the shoots were dried at 70 °C for 48 h. The dry samples of the shoots were ground, homogenized and combusted in the presence of perchloric and nitric acid (6:1). The phosphorus concentration was determined spectrophotometrically (Thermo Scientific™ Evolution™ 60S UV-Visible Spectrophotometer) using the molybdovanadate method [37]. The calcium, magnesium, iron, manganese and zinc content were determined by atomic absorption spectroscopy (Thermo Scientific - SOLAAR M Series AA Spectrometer) [37].

### 5.4. Root System Measurements

The roots were washed and cleaned from the substrate and scanned with an Epson Perfection V700 scanner (Seiko Epson Corporation, Nagano Japan). To determine number of root tips, the root width, depth, total root length, surface area, volume and average root diameter, the root system images were analysed using WinRHIZO Pro (Regent Instruments Inc., Quebec, QC, Canada).

### 5.5. AM Root Colonization Analysis

To determine the extent of the AM root colonization, the 30 youngest root parts (1 cm) per plant were cleared with hot 10% KOH and acidified with 1 mol L^−1^ HCl. The root parts were stained with 0.05% Trypan Blue in lactoglycerol [40]. The frequency of mycorrhizas in the root system (F%) and the extent of the root cortex colonization (M%) were calculated according to [41], using the MYCOCALC program [42].

### 5.6. Mycorrhizal Responsiveness

The responsiveness of each cultivar to AM colonization was estimated by the calculation of:(1)The improvement in shoot biomass as the percent difference in the shoot dry weight between the AM and non-mycorrhizal plants [22],(2)The improvement of P nutrition, as the percent difference in the shoot P concentration between the AM and non-mycorrhizal plants [43](3)The improvement in root growth as the percent difference in the total root length.

The responsiveness was calculated using the following equation:Responsiveness = ((trait value of AMI plants − trait value of AMC plants)/trait value of AMC plants) × 100,(5)

### 5.7. Data Analysis

The analysis of variance (ANOVA) in PROC MIXED was performed using the SAS system for Windows [44]. The cultivars (*n* = 4) and AM treatments (*n* = 2) were set as fixed effects, and the replications (*n* = 4) as random effects. Multiple comparison procedures were performed and pairwise differences were computed for the fixed effects and their interactions using the Tukey–Kramer test (P ≤ 0.05). The normality was tested by the Shapiro–Wilks test and the homoscedasticity by assessing the plot of residuals against fitted values. The data are presented as means ± standard deviation (SD).

## Figures and Tables

**Figure 1 plants-09-00014-f001:**
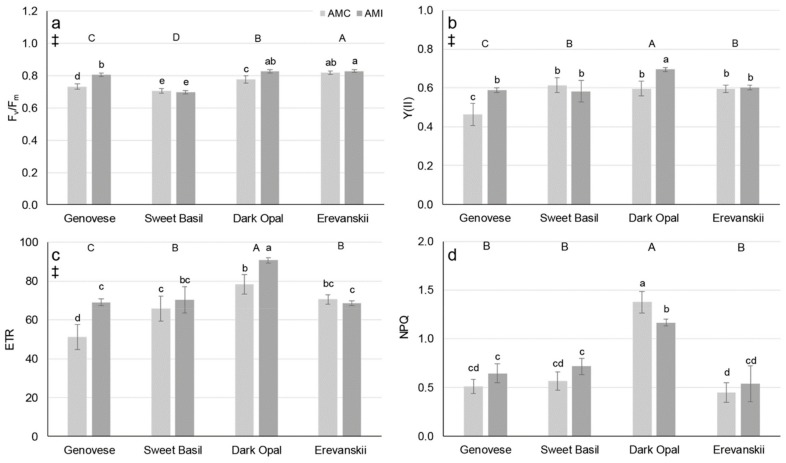
Chlorophyll fluorescence parameters (**a**) maximal quantum yield of PS 2 (Fv/Fm), (**b**) actual quantum yield of PS 2 (Y(II)), (**c**) apparent electron transport rate (ETR), and (**d**) non-photochemical quenching (NPQ) of basil cultivars non-inoculated (AMC) or inoculated (AMI) with arbuscular mycorrhizal fungi (*Rhizophagus irregularis*). Mean values are shown by histograms, error bars represent standard deviation. ‡ denotes significant (P ≤ 0.05) differences between AMI and AMC, different uppercase letters indicate significant (P ≤ 0.05) differences among cultivars, different lowercase letters associated with histograms, indicate significant (P ≤ 0.05) pairwise differences for Cultivar × AM treatment interaction.

**Figure 2 plants-09-00014-f002:**
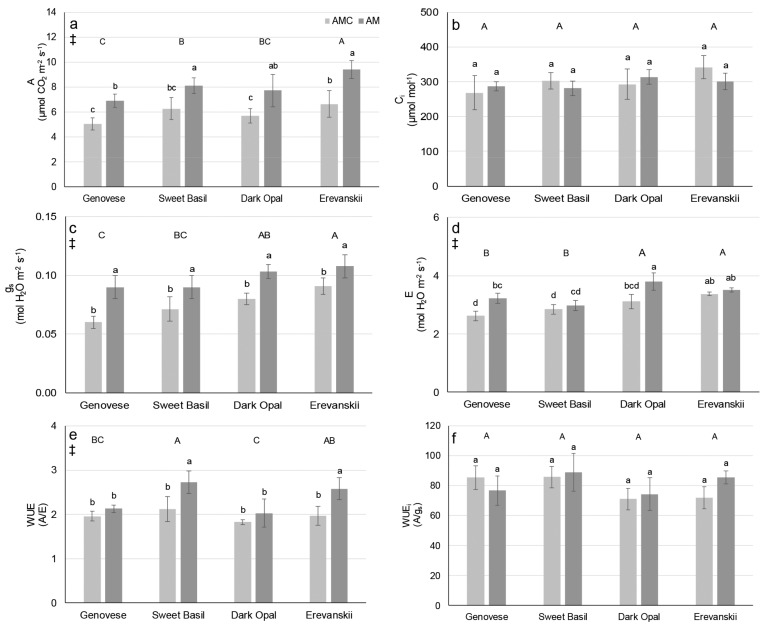
Gas-exchange parameters (**a**) net photosynthetic rate (A), (**b**) mesophyll CO_2_ concentration (C_i_), (**c**) stomatal conductance (g_s_), (**d**) transpiration rate (E), (**e**) water-use efficiency (WUE), and (**f**) intrinsic water use efficiency (WUE_i_) of basil cultivars non-inoculated (AMC) or inoculated (AMI) with arbuscular mycorrhizal fungi (*Rhizophagus irregularis*). Mean values are shown by histograms, error bars represent standard deviation. ‡ denotes significant (P ≤ 0.05) differences between AMI and AMC, different uppercase letters indicate significant (P ≤ 0.05) differences among cultivars, different lowercase letters associated with histograms, indicate significant (P ≤ 0.05) pairwise differences for Cultivar × AM treatment interaction.

**Figure 3 plants-09-00014-f003:**
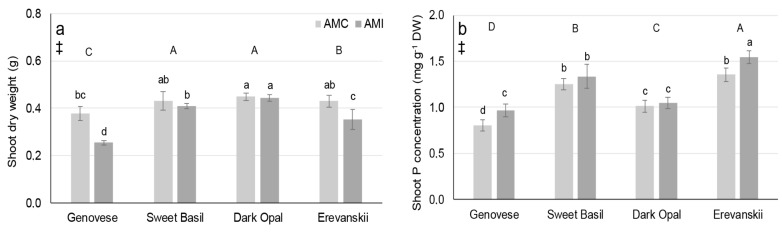
Shoot biomass and phosphorus concentration (**a**) shoot dry weight (**b**) and shoot phosphorus (P) concentration (mg g^−1^ DW) of basil cultivars non-inoculated (AMC) or inoculated (AMI) with arbuscular mycorrhizal fungi (*Rhizophagus irregularis*). Mean values are shown by histograms, error bars represent standard deviation. ‡ denotes significant (P ≤ 0.05) differences between AMI and AMC, different uppercase letters indicate significant (P ≤ 0.05) differences among cultivars, different lowercase letters associated with histograms, indicate significant (P ≤ 0.05) pairwise differences for Cultivar × AM treatment interaction.

**Figure 4 plants-09-00014-f004:**
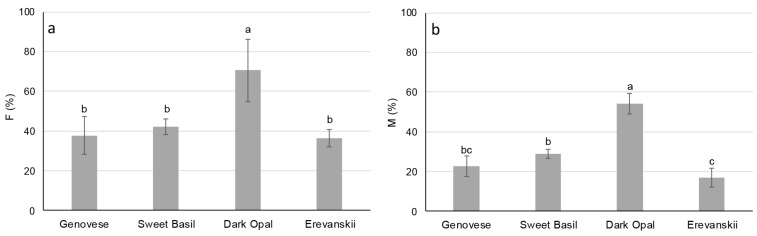
Arbuscular mycorrhiza root colonization (**a**) frequency of mycorrhizas in root system (F%) and (**b**) the extent of root cortex colonization (M%) of basil cultivars by arbuscular mycorrhizal fungi (*Rhizophagus irregularis*). Mean values are shown by histograms, error bars represent standard deviations, and different small case letters associated with histograms, indicate significant differences (P ≤ 0.05) among basil cultivars.

**Figure 5 plants-09-00014-f005:**
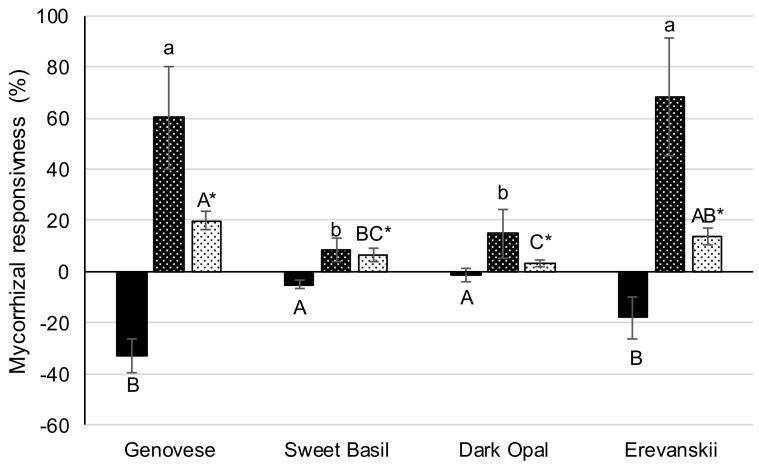
Mycorrhizal responsiveness of different basil cultivars. Responsiveness in terms of per-cent difference in shoot biomass increase █, root length ▓, and shoot P concentration ▒. Each value represents the mean of four replicates. Mean values are shown by histograms, error bars represent standard deviations, and different lower-case letters indicate differences (P ≤ 0.05) in root length responsiveness; upper-case letters indicate differences (P ≤ 0.05) in shoot biomass responsiveness and upper-case letters with asterisk indicate differences (P ≤ 0.05) in P content responsiveness among basil cultivars.

**Table 1 plants-09-00014-t001:** Mean values (± standard deviation) and analysis of variance shoot mineral content of basil cultivars non-inoculated (AMC) and inoculated (AMI) with arbuscular mycorrhizas (*Rhizophagus irregularis*).

Cultivar	AM inoculation	Ca (%)	Mg (%)	Fe (mg g^−1^ DW)	Mn (mg g^−1^ DW)	Zn (mg g^−1^ DW)
‘Genovese’	AMC	1.29 ± 0.30 ^b^	0.45 ± 0.06 ^a^	86.2 ± 7.9 ^ab^	54.1 ± 4.9 ^a^	22.7 ± 1.5 ^b^
	AMI	1.16 ± 0.18 ^b^	0.43 ± 0.06 ^a^	82.3 ± 8.2 ^ab^	52.5 ± 6.5 ^a^	20.5 ± 1.5 ^b^
‘Sweet Basil’	AMC	1.26 ± 0.17 ^b^	0.44 ± 0.09 ^a^	83.0 ± 6.1 ^ab^	49.1 ± 3.3 ^a^	20.7 ± 1.4 ^b^
	AMI	1.12 ± 0.17 ^b^	0.45 ± 0.08 ^a^	83.0 ± 7.0 ^ab^	46.8 ± 3.2 ^a^	22.3 ± 2.7 ^b^
‘Dark Opal’	AMC	1.08 ± 0.23 ^b^	0.30 ± 0.06 ^b^	93.6 ± 9.3 ^a^	47.6 ± 4.8 ^a^	31.5 ± 3.2 ^a^
	AMI	1.23 ± 0.22 ^b^	0.31 ± 0.05 ^b^	90.8 ± 10.2 ^a^	49.9 ± 7.0 ^a^	32.5 ± 2.3 ^a^
‘Erevanskii’	AMC	1.58 ± 0.15 ^a^	0.34 ± 0.09 ^b^	79.9 ± 9.7 ^b^	45.4 ± 6.2 ^a^	21.4 ± 1.5 ^b^
	AMI	1.48 ± 0.15 ^a^	0.32 ± 0.05 ^b^	75.3 ± 9.2 ^b^	43.3 ± 6.3 ^a^	20.3 ± 2.2 ^b^
Source of Variation		Significance level (P)				
Cultivar (C)		<0.01	<0.01	<0.05	n.s.	<0.001
Inoculation Treatment (M)		n.s.	n.s.	n.s.	n.s.	n.s.
C × M		n.s.	n.s.	n.s.	n.s.	n.s

For the analysis of variance (ANOVA), n.s. denotes not significant. Different lowercase letters indicate pairwise differences (P ≤ 0.05).

**Table 2 plants-09-00014-t002:** Mean values (± standard deviation) and analysis of variance for root morphological traits of the basil cultivars non-inoculated (AMC) and inoculated (AMI) with arbuscular mycorrhizas (*Rhizophagus irregularis*).

Cultivar	AM Inoculation	Depth (cm)	Width (cm)	Length (cm)	Surface Area (cm^2^)	Volume (cm^3^)	Diameter (mm)	Number of Tips
‘Genovese’	AMC	13.3 ± 0.6 ^b^	8.1 ± 0.7 ^ab^	187.6 ± 23.1 ^b^	6.0 ± 0.4 ^b^	0.18 ± 0.018 ^bc^	0.32 ± 0.025 ^ab^	111.7 ± 14.1 ^b^
	AMI	21.3 ± 1.2 ^a^	10.4 ± 2.1 ^ab^	334.4 ± 37.1 ^a^	11.3 ± 1.7 ^a^	0.32 ± 0.029 ^a^	0.27 ± 0.014 ^b^	317.5 ± 88.7 ^a^
‘Sweet Basil’	AMC	12.1 ± 1.0 ^b^	5.4 ± 1.1 ^c^	108.7 ± 19.3 ^b^	3.9 ± 1.0 ^b^	0.11 ± 0.037 ^cd^	0.36 ± 0.001 ^a^	110.0 ± 39.8 ^b^
	AMI	13.1 ± 1.2 ^b^	6.6 ± 1.2 ^bc^	114.1 ± 45.7 ^b^	3.8 ± 0.6 ^b^	0.11 ± 0.040 ^cd^	0.33 ± 0.016 ^a^	102.8 ± 31.5 ^b^
‘Dark Opal’	AMC	10.7 ± 1.1 ^c^	7.3 ± 2.5 ^bc^	143.0 ± 53.1 ^b^	3.9 ± 0.5 ^b^	0.08 ± 0.010 ^d^	0.27 ± 0.006 ^b^	98.3 ± 10.62 ^b^
	AMI	12.3 ± 2.0 ^b^	7.5 ± 0.7 ^bc^	186.1 ± 18.1 ^b^	5.3 ± 1.5 ^b^	0.12 ± 0.017 ^cd^	0.28 ± 0.014 ^b^	115.8 ± 18.7 ^b^
‘Erevanskii’	AMC	12.6 ± 1.2 ^b^	8.1 ± 1.1 ^abc^	197.2 ± 24.2 ^b^	4.7 ± 0.5 ^b^	0.12 ± 0.011 ^cd^	0.26 ± 0.011 ^bc^	140.0 ± 60.0 ^b^
	AMI	21.2 ± 2.4 ^a^	12.1 ± 1.5 ^a^	388.2 ± 33.1 ^a^	10.1 ± 0.9 ^a^	0.21 ± 0.019 ^b^	0.23 ± 0.004 ^c^	331.2 ± 86.5 ^a^
Source of Variation		Significance level (P)						
Cultivar (C)		<0.001	<0.01	<0.001	<0.01	<0.001	<0.001	<0.001
Inoculation Treatment (M)		<0.001	<0.05	<0.001	<0.01	<0.001	<0.05	<0.01
C × M		<0.001	n.s.	<0.05	<0.05	<0.05	n.s.	<0.01

For the ANOVA, n.s. denotes not significant. Different lowercase letters indicate pairwise differences (P ≤ 0.05) for Cultivar × AM treatment interaction.

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
