# Peer review of "Physiological Responses of Basil (*Ocimum Basilicum* L.) Cultivars to *Rhizophagus Irregularis* Inoculation under Low Phosphorus Availability"

_plants, 2019, doi:10.3390/plants9010014_

Round 1

Reviewer 1 Report

This manuscript is interesting and quite well written. I have to commend the authors for a comprehensive evaluation of the described parameters. On the other hand, I have some comments:

Discussion

- Text in this chapter needs a check-up of figure numbers because some of them do not correspond with the right pictures, which makes the reading and understanding messy (line 201 - correct Figure 4a to Figure 3a; line 208 - correct Figure 4a to Figure 3).

Materials and Methods

- There is definitely no description of vermiculite composition. This information is crucial because the whole article deals with conditions of low phosphorus availability for plants. At least the basic chemical and physical properties must be mentioned.

- A small number of experimental members is the shortcoming of this study. It is risky to make these conclusions based on the response of four plants per treatment.

Author Response

Dear reviewer

First, we would like to thank you for the time devoted to the review and for the fruitful suggestions that have improved the quality of our manuscript.

We have used track changes, thus all modifications within the manuscript are highlighted.

Best regards

Authors

Reviewer 2 Report

The results in manuscript “Physiological Responses of Basil (Ocimum basilicum L.) Cultivars to Rhizophagus irregularis Inoculation Under Low Phosphorus Availability” are interesting but the manuscript have to be corrected according to following comments:

Too small amounts of plants (only 4) were investigated. This is not the science.

The sentences such as - “Significantly higher or lower differences among basil cultivars ...“ are correct only in the case when the differences were significantly higher or lower in all cultivar, not only one or two. In addition, the next sentence is about only significant differences of one or two cultivars. See lines: 83-86, 98-101, 113-117, 121-125, 136-138. The words - “significantly“ or “significant“ – must be deleted from lines: 83, 84, 98, 113, 122, 136.

The authors discussed about increased C and N partitionig into root system. Please, add this information to the table (amount of C, N in roots and also in shoots).

The information about amount of P in roots is need to add to figure/table and also simple calculations of common indicators used in evaluation of plant tolerance to stress treatment, growth tolerance index (GTI, on a basis of shoot/root biomass) and bioconcentration factors (BCF) for nutrient elements, are need to add.  Why authors did not calculate the amount of P in one plants in shoot and roots? Please, add this information.

It is interesting that  shoot dry weight decreased in Genovese and Erevanski, but the net photosynthetic rate, .... increased (Fig. 2). How do authors explain this? Why did not authors determined also the amount/concentration of other elements (Mn, Fe, Mg, ...) which are connected with photosynthesis.

The text in lines 75-79 and in lines 82-86 is identical. Please, delete the one.

Reviewer 3 Report

An interesting subject, suitable for Plants, and a well executed experiment, the text is sufficiently clean and clear. The title accurately describes the paper and the abstract is clear and correct in summarizing the paper. The keywords are satisfactory and the tables are necessary. A careful check have to be made on references formatting.

I only suggest to authors to include a section "Conclusions". A good form would be to point out the obtained results allowing to determine what results according to the Authors are priority and which are of secondary importance.

I believe that, after considering these comments, the article submitted for the evaluation fully deserves publication in the scientific journal "Plants".

Author Response

First, we would like to thank the reviewer for the time devoted to the review and for the suggestions that have improved the quality of our manuscript.

We have used track changes, thus all modifications within the manuscript are highlighted.

Best regards

Authors

Round 2

Reviewer 2 Report

The results in manuscript “Physiological Responses of Basil (Ocimum basilicum L.) Cultivars to Rhizophagus irregularis Inoculation Under Low Phosphorus Availability” are only preliminary, so the manuscript is not acceptable for publishing in “Plants”.

The major reservations are:

Point 1 - Too small amounts of plants (only 4) were investigated. This is not the science. The results are no considerable and are only preliminary. The response of authors is inadequate.

Point 2 – The science is exactly and the differences are significant only in the case if the pairwise differences are significant. Only the significant differences must be mentioned in the text. The result that the ANOVA show p<0.05 is not satisfactory. The pairwise differences are important. The authors did not correct the text. For example - Lines 76-79: Significantly higher or lower differences are only in Genovese and Dark Opal. The sentences must be deleted and the correct sentence is: Significantly higher Fv/Fm, Y(II) and ETR were found in AMI plants compared to control plants only in ‘Genovese’ and ‘Dark Opal’, and significantly lower NPQ in ‘Dark Opal’ (Figure 1). The same is in lines 89-93, 104-124, 138-146.

Points 3 and 4 - The response of authors is inadequate. The authors can realize the additional experiments to obtain the necessary results. The discussion will be rewritten according the new results. In this version, the discussion is only poor.

Point 5 – There are no differences between the AMC and AMI plants in concentration of Ca, Mg, Fe, Mn, Zn. The information or explanation, why shoot dry weight decreased in Genovese and Erevanski, but the net photosynthetic rate, .... increased, is still not answered.

Author Response

Dear reviewer

Thank you for the time devoted to the review and for the suggestions. We have tried to acknowledge all suggestions and comment which you have suggested, unfortunately some things couldn’t be done within a correction period.  Below you can find our response to your comments.

Point 1.

Point 1 - Too small amounts of plants (only 4) were investigated. This is not the science. The results are no considerable and are only preliminary. The response of authors is inadequate.

Response 1

We are aware of this limitation (issue).

In addition to previous response, we have added that these results can be considered as a preliminary, and that further experiments are needed to confirm these results.

Please see lines:

LINES 279-280

Point 2.

Point 2 – The science is exactly and the differences are significant only in the case if the pairwise differences are significant. Only the significant differences must be mentioned in the text. The result that the ANOVA show p<0.05 is not satisfactory. The pairwise differences are important. The authors did not correct the text. For example - Lines 76-79: Significantly higher or lower differences are only in Genovese and Dark Opal. The sentences must be deleted and the correct sentence is: Significantly higher Fv/Fm, Y(II) and ETR were found in AMI plants compared to control plants only in ‘Genovese’ and ‘Dark Opal’, and significantly lower NPQ in ‘Dark Opal’ (Figure 1). The same is in lines 89-93, 104-124, 138-146.

Response 2

Acknowledged, all results were rewritten as suggested.

Please see LINES: 76-82; 93-98; 109-122; 143-152.

Point 3. and 4

Points 3 and 4 - The response of authors is inadequate. The authors can realize the additional experiments to obtain the necessary results. The discussion will be rewritten according the new results. In this version, the discussion is only poor.

Response 3 and 4

We are aware of the small sample size and acknowledged it within the manuscript (please see the response 1).  Root samples were used for analysis of the frequency of mycorrhizas in the root system (F%) and the extent of the root cortex colonization (M%), and due to the sample preparation, these root samples were not usable for further analysis. We agree that suggested analysis would be of great value for the discussion and confirmation of these findings. Further experiments are always the option, however it would take  several months to conduct the experiment and analyse all traits again, which is not doable within the period provided for corrections of the manuscript. 

Point 5.

Point 5 – There are no differences between the AMC and AMI plants in concentration of Ca, Mg, Fe, Mn, Zn. The information or explanation, why shoot dry weight decreased in Genovese and Erevanski, but the net photosynthetic rate, .... increased, is still not answered.

Response 5.

Few lines of discussions are included.

Please see LINES: 264-269

Best regards

Authors
